# Functional specialization of the human posterior parietal cortex in visually and proprioceptively driven reaching corrections

Riccardo Brandolani [1,2], Claudio Galletti[1], Davide Di Gloria[3], Patrizia Fattori [1,4] & Rossella Breveglieri [1,4] ✉

While online adjustments during reaching are essential for interacting with our dynamic environment, the specialized contributions of subregions of the posterior parietal cortex (PPC) remain unclear. In this study, we investigate the distinct roles of human medial PPC areas V6A (hV6A) and PEc (hPEc) in mediating online reaching corrections elicited by visual and proprioceptive perturbations. Here we deliver online repetitive transcranial magnetic stimulation (rTMS) during the early stages of reaching corrections triggered by an unexpected shift of the visual target or by the application of an external force to the wrist. Our findings reveal that rTMS over hV6A selectively impairs the ability to correct reach trajectories for visual perturbations, whereas stimulation of hPEc interferes only with proprioceptively driven corrections. These findings confirm the critical role of hV6A in processing visual feedback, demonstrate the causal involvement of hPEc in integrating proprioceptive information to guide motor adjustments, and show how the PPC selectively engages specialized neural circuits to adapt motor control strategies according to the sensory nature of the reaching perturbation.

The ability to swiftly adjust our movements in an ever-changing environment is essential in everyday life. Understanding how this skill operates and identifying the brain regions responsible for its control are crucial for advancements in fields such as neurorehabilitation[1,2], neural prosthetics[3], and fundamental neuroscience research[4–6]. It is well established that vision and proprioception contribute both uniquely and interactively to the planning, execution, and adjustments of reaching movements, and that the posterior parietal cortex (PPC) plays a crucial role in integrating these sensory inputs to optimize motor control[4,6–16]. The PPC contains areas which are functionally specialized and have different sensory-related modulations. In its medial part, PPC hosts area V6A posteriorly (monkey V6A[17], the human homologue hV6A[18]) and area PEc immediately anterior to V6A (monkey PEc[19], the human homologue hPEc[20]). Both areas contain visual and somatosensory cells, with a higher incidence of visual cells in V6A and of somatosensory cells in PEc[21–25]. These areas are also involved in arm movements[22,24,26], with hPEc activated also during locomotion[26–28].

The causal role of brain areas during reaching corrections can be investigated using transcranial magnetic stimulation (TMS) with high temporal resolution. According to recent TMS studies, hV6A is causally involved in reaching corrections elicited by shifts of the visual target[5] and during hand pre-shaping of a reach-to-grasp action perturbed by the application of an external force[29]. Nevertheless, it is still unknown whether hV6A is also involved in movement corrections in a spatial reaching task following proprioceptive perturbations. Regarding hPEc, our understanding of its causal role in reaching perturbations is even more incomplete, because, to the best of our knowledge, this has never been tested.

To fill this gap in knowledge and to understand functional specialization within the human medial PPC, in this study we investigated whether and how areas hV6A and hPEc contribute to reaching corrections triggered by perturbations of visual target position, arm position, or both. To address this point, we designed a SHAM-controlled TMS experiment where repetitive TMS (rTMS) was delivered over hV6A or hPEc (in addition to control

[1]Department of Biomedical and Neuromotor Sciences, University of Bologna, Piazza di Porta San Donato, 2, 40126 Bologna, Italy. [2]University of Camerino, Center for Neuroscience, 62032 Camerino, Italy. [3]STAM s.r.l., via Lorenzo Pareto 8AR, 16129 Genova, Italy. [4]Interdepartmental center for industrial research-Aerospace, University of Bologna, Via Baldassarre Carnaccini 12, 47121 Forlì FC, Italy. ✉e-mail: rossella.breveglieri@unibo.it

sites) during the early phases of unperturbed and perturbed reaching movements. We found that hV6A plays a specific role in early corrections of reaching movements in response to spatial shifts of the visual target, but is not involved in early reaching corrections following the interference of an external force on the arm. In contrast, we showed for the first time that hPEc is only involved in corrections following perturbations of arm position. These novel results strongly suggest distinct functions of PPC subregions in motor control.

## Results

We conducted two experiments. In the first one (Fig. 1A), online repetitive transcranial magnetic stimulation (rTMS, 3 pulses during reaching execution) was applied over hV6A as the area of interest, and over V1/V2 as control area, including also a SHAM condition. In the second experiment (Fig. 1B), hPEc was the area of interest, IPS served as the control area, and a SHAM condition was included too.

The two experiments were identical in design, differing only in the brain regions targeted for stimulation and in the participants involved. Each trial began with participants pressing a home button (Fig. 2A, B), which triggered the appearance of a central green fixation point on the screen. Participants were instructed to maintain their gaze on this point throughout the trial. After a variable delay of 1.3–1.5 seconds, a gray target appeared to the right of the fixation point, either at 11° (near) or 20° (far) of eccentricity, serving as the go signal for initiating a reaching movement. Participants were asked to reach toward the target with their index finger. The task included two types of trials: unperturbed (80%) and perturbed (20%). In unperturbed trials, the target remained stationary (40% near, 40% far). In perturbed trials, two types of rightward corrections were required. In 10% of trials, a visual perturbation occurred: the target initially appeared in the near position and shifted to the far position 50 ms after movement onset. In another 10% of trials, a proprioceptive perturbation was introduced: the target remained in the far position, but a mechanical force pulled the participant's wrist leftward 50 ms after movement onset, requiring a rightward correction to reach the target. In both perturbation types, the required correction was in the same direction. Three TMS pulses were delivered during movement execution in all trials, with the first pulse occurring 50 ms

after movement onset (Fig. 2C). In each experiment, unperturbed, visually perturbed and proprioceptively perturbed trials were run within subjects and within blocks, to avoid participants forming modality-specific expectations in the different phases of the experiment.

### Role of hV6A in reaching corrections to visual perturbations

To study the role of parietal areas in reaching corrections triggered by the shifting of a visual target, we evaluated whether the stimulation affected the correction ability of the participants, measured as the Euclidean Distance between perturbed and unperturbed trajectories (ED, see Methods and Fig. 6). If the ED profile over time is altered with TMS, a causal role of the brain region in reaching corrections can be suggested. The results of the two repeated measures analysis of variance (ANOVA, one for each experiment, the first one stimulating hV6A and V1/V2, while the second one hPEc and IPS), their significance and effect sizes are summarized in Table 1. In each ANOVA, we tested the effects of within-subject factors TMS (3 levels, SHAM, V1/V2, hV6A for experiment 1 and SHAM, IPS, hPEc for experiment 2) and Bin (10 levels, bin 1-10, each one representing 10% of the movement time). The specific time bins where significant effects were observed are reported in Fig. 3 for readers' convenience. In all the other bins, all $p > 0.05$. Among all the tested areas (region of interest (ROI, hV6A and hPEc) and active control sites, IPS, V1/V2), only hV6A stimulation impaired the corrections of participants during reaching reprogramming triggered by the shift of the position of the visual target, in all the tested kinematic markers (compare the blue traces with the others in Fig. 3A and see Table 1). Specifically, hV6A stimulation resulted in a significantly lower ED between perturbed and unperturbed trajectories of the index finger compared to SHAM (all significant $p < 0.03$, Fig. 3A) and to V1/V2 (all significant $p < 0.03$). The wrist trajectories were influenced similarly: hV6A stimulation produced a lower ED than SHAM (all significant $p < 0.03$) and lower than V1/V2 (all significant $p < 0.05$). The same holds true for the forearm, where hV6A stimulation led to a lower ED compared to SHAM (all significant $p < 0.02$) and compared to V1/V2 (all significant $p < 0.02$). The smaller ED between hV6A and SHAM is informative about impairments in reaching corrections, in particular, the perturbed trajectories after hV6A stimulation stayed closer to the unperturbed ones than SHAM, causing a delay in the reaching correction dynamics. Regarding the upper arm, hV6A stimulation showed a higher ED than SHAM at the end of the movement ($p < 0.02$), and than V1/V2 (all significant $p < 0.02$). The EDs of SHAM and V1/V2 were not significantly different across all kinematic markers (all $p > 0.11$). Neither the stimulation of hPEc nor of IPS resulted in a change of the correction skills (Fig. 3B, all $p > 0.1$).

Overall, our results clearly show that hV6A stimulation selectively impaired the dynamics of reaching corrections triggered by shifts of the target position. This effect was not present for hPEc, V1/V2, and IPS stimulation. The impairment after hV6A stimulation was evident across all kinematic markers, although with different timings. Specifically, changes in the correction ability of the index, wrist and forearm were observed during the mid-phase of the movement, indicating a greater difficulty in adjusting the reaching trajectory to the new target position following a visual target shift. In contrast, the effects on the upper arm were found in the later stages of the movement, possibly due to postural compensations impaired by TMS over hV6A. These results support the hypothesis that hV6A, but not the other tested areas, is causally involved in the online reprogramming of reaching in response to target position changes.

### Effects of TMS over several regions in response to proprioceptive perturbations

To study the specific role of the medial posterior parietal areas in proprioceptive perturbations, we looked for correction changes during proprioceptive perturbations following TMS. Here, the higher ED between each stimulation condition and SHAM is informative about correction impairments (Fig. 6B). Specifically, the perturbed trajectories after stimulation shifted more than SHAM from the unperturbed ones, causing a delay in reaching correction dynamics (see Table 2 for the two repeated measures

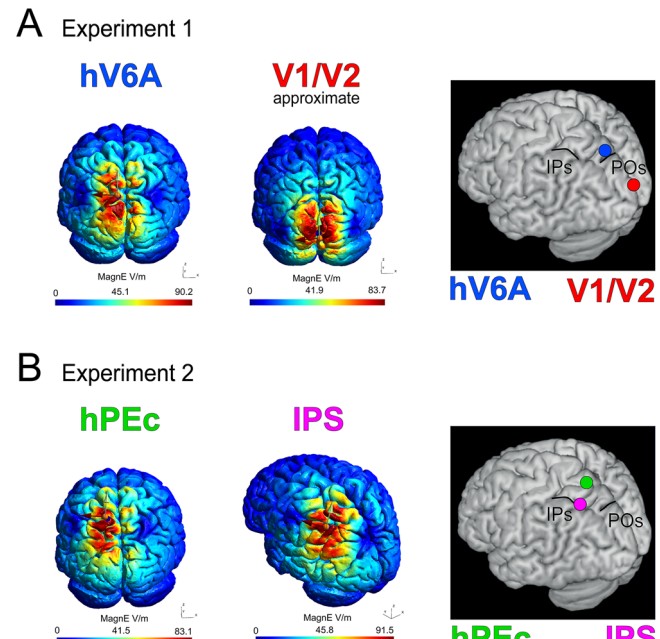

**A** Experiment 1

hV6A    V1/V2
approximate

MagnE V/m    MagnE V/m
0    45.1    90.2    0    41.9    83.7

IPs    POs

hV6A    V1/V2

**B** Experiment 2

hPEc    IPS

MagnE V/m    MagnE V/m
0    41.5    83.1    0    45.8    91.5

IPs    POs

hPEc    IPS

**Fig. 1 | Localization of the TMS brain sites.** TMS was delivered over four cortical regions across experiment 1 (**A**) and experiment 2 (**B**): hV6A, V1/V2, hPEc, and IPS. Target sites were identified using neuronavigation and further validated through finite-element modeling of the induced electric field (E-field).

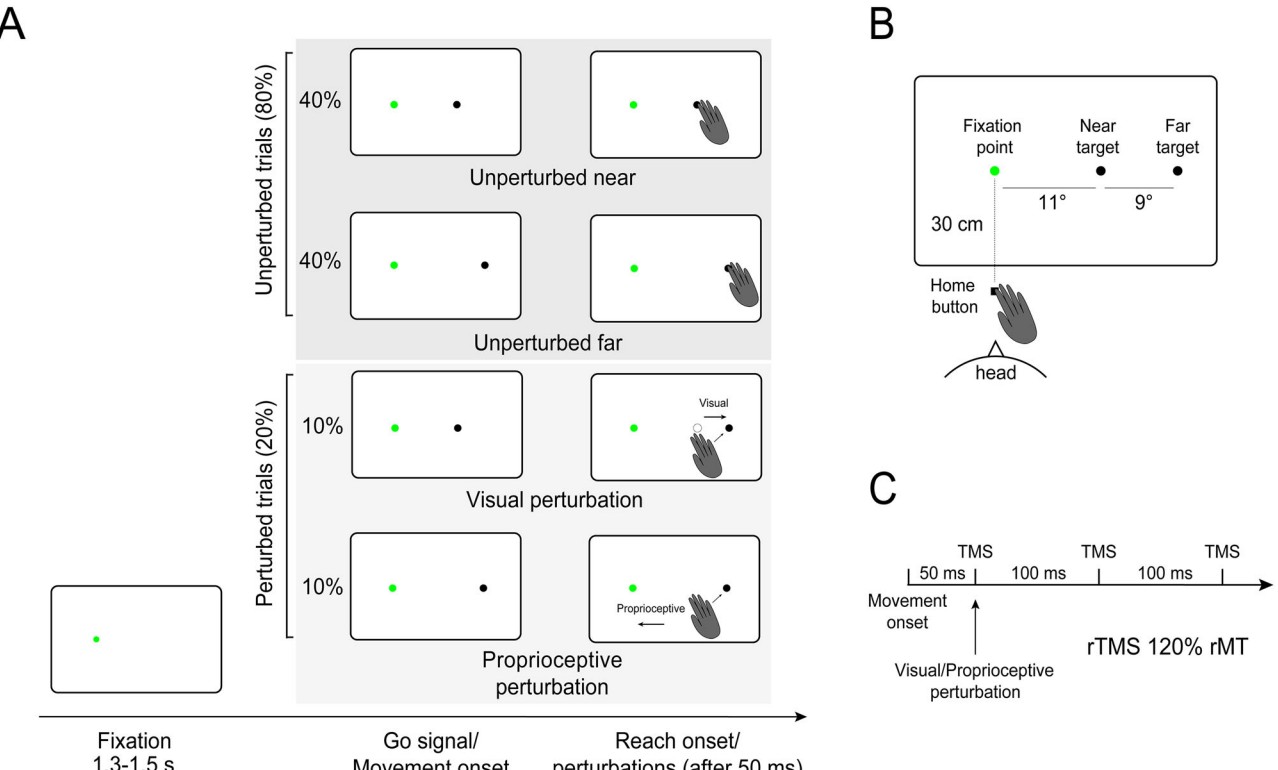

**Fig. 2 | Experimental paradigm. A** Experimental design and trial sequence: each experiment began with a home button press that displayed a green fixation dot on a touchscreen. After 1.3–1.5 seconds, a gray target appeared either 11° or 20° to the right, prompting participants to reach toward it. Most trials (80%) were unperturbed, while 20% were perturbed trials, where the target either shifted visually or the participant's arm was pulled, requiring online movement adjustments in the same direction. **B** Spatial arrangement of the stimuli. **C** rTMS protocol and timing; rMT = resting motor threshold.

ANOVA effects with TMS and bin as within-subject factors). The specific time bins where significant effects were observed are reported in Fig. 4 for readers' convenience. In all the other bins, all $p > 0.05$.

The stimulation of PEc and of the control site IPS caused effects on all the markers except the most distal one: PEc stimulation caused higher ED of the wrist than SHAM late in the movement ($p < 0.01$) and than IPS (all significant $p < 0.04$). Additionally, IPS stimulation produced an earlier, higher ED than SHAM (all significant $p < 0.03$) and than hPEc ($p < 0.04$, Fig. 4B).

The same pattern and timing were observed for the forearm and the upper arm: the stimulation of hPEc produced higher ED of the forearm than both SHAM and IPS (all significant $p < 0.03$), while the stimulation of IPS resulted in a higher ED than of hPEc halfway through the movement ($p < 0.03$). The stimulation of hPEc caused higher values of ED of the upper arm than SHAM ($p < 0.02$) and IPS (all significant $p < 0.05$), whereas IPS stimulation produced a higher ED than SHAM (all significant $p < 0.03$) and higher than hPEc ($p < 0.05$).

The stimulation of the control occipital site impaired the correction capabilities of the most distal segment, the index finger, during proprioceptive perturbations compared to SHAM (all significant $p < 0.03$) and compared to hV6A (all significant $p < 0.03$, Fig. 4A, left). No other significant effects were detected (all $p > 0.1$).

Overall, while hPEc and IPS stimulation had no impact on reaching corrections during visual perturbations, strong and consistent effects across multiple kinematic markers, except the most distal one, were observed during proprioceptive perturbation. Notably, the timing of these impairments was different: IPS stimulation primarily affects reaching corrections halfway through the movement, whereas hPEc stimulation induces later changes. Interestingly, no effect was observed on the index finger, possibly due to a compensating mechanism that allowed participants to maintain the index finger in a position that ensured high accuracy. Surprisingly, we also observed an effect of V1/V2 stimulation on the corrections operated by the index finger. hV6A stimulation did not affect reaching corrections following proprioceptive perturbations.

EMG recordings (Fig. S1-2), movement times (Fig. S3) and reaction time analyses validated our task design and are reported in the Supplementary material. Stimulation of parietal and occipital areas had no effect on unperturbed reaching movements (Fig. S4, Table S1). No significant effects on reaching accuracy were reported in either experiment (see Supplementary material).

## Discussion

In this study, we examined the distinct roles of parietal areas hV6A and hPEc in visually guided reaching corrections. Our results show that hV6A is critically involved in the early corrections to visual reaching perturbations following visual target shifts, but not in corrections to proprioceptive perturbations, confirming its established role in visuomotor integration. In contrast, hPEc was selectively recruited during later-stage corrections driven by the application of external forces to the arm, consistent with its robust somatosensory responsiveness reported in electrophysiological studies in non-human primates. These findings suggest that the PPC selectively engages specialized neural circuits to adapt motor control strategies according to the sensory nature of the perturbation.

### hV6A is involved in the early updating of the representation of visual space during reaching corrections

These findings are consistent with previous research, which identified specific functions for the different regions of the PPC. Area V6A is primarily recognized as a visuomotor area in both humans and monkeys[16,30,31]. The present results, showing a role for hV6A in reaching corrections triggered by the shift of the spatial position of the visual target, well agree with this role. In this study, we also found that hV6A is not involved in early corrections of a visually guided reaching after a perturbation caused by an application of an

**Table 1 | ANOVA results for the ED during visual perturbations**

| | VISUAL PERTURBATION | | | | | | | | | | |
| --- | --- | --- | --- | --- | --- | --- | --- | --- | --- | --- | --- |
| | EXPERIMENT 1 | | | | | | EXPERIMENT 2 | | | | |
| INDEX | SS | DOF | MS | F | p | Partial η² | SS | DOF | MS | F | p | Partial η² |
| TMS | 738.4 | 2 | 369.2 | 2.48 | 0.10 | 0.14 | 205.6 | 2 | 102.8 | 0.54 | 0.59 | 0.04 |
| Error | 4460.6 | 30 | 148.7 | | | | 4944.3 | 26 | 190.2 | | | |
| Bin | 232776.7 | 9 | 25864.1 | 90.90 | **<0.001** | 0.86 | 154155.1 | 9 | 17128.3 | 47.25 | **<0.001** | 0.78 |
| Error | 38411.2 | 135 | 284.5 | | | | 42411.9 | 117 | 362.5 | | | |
| TMS*bin | 1017.2 | 18 | 56.5 | 2.55 | **<0.001** | 0.15 | 144.9 | 18 | 8 | 0.30 | 0.99 | 0.02 |
| Error | 5990.6 | 270 | 22.2 | | | | 6275.7 | 234 | 26.8 | | | |
| WRIST | SS | DOF | MS | F | p | Partial η² | SS | DOF | MS | F | p | Partial η² |
| TMS | 195.5 | 2 | 97.7 | 0.91 | 0.41 | 0.06 | 113.81 | 2 | 56.9 | 0.80 | 0.46 | 0.06 |
| Error | 3225.4 | 30 | 107.5 | | | | 1854.52 | 26 | 71.33 | | | |
| Bin | 78989.9 | 9 | 8776.7 | 69.99 | **<0.001** | 0.82 | 33570.15 | 9 | 3730.02 | 23.35 | **<0.001** | 0.64 |
| Error | 16928.0 | 135 | 125.4 | | | | 18687.82 | 117 | 159.72 | | | |
| TMS*bin | 720.1 | 18 | 40 | 3.90 | **<0.001** | 0.21 | 83.94 | 18 | 4.66 | 0.32 | 0.99 | 0.02 |
| Error | 2764.0 | 270 | 10.2 | | | | 3378.52 | 234 | 14.44 | | | |
| FOREARM | SS | DOF | MS | F | p | Partial η² | SS | DOF | MS | F | p | Partial η² |
| TMS | 286.98 | 2 | 143.49 | 1.87 | 0.17 | 0.11 | 75.12 | 2 | 37.56 | 0.50 | 0.61 | 0.04 |
| Error | 2303.4 | 30 | 76.78 | | | | 1948.16 | 26 | 74.93 | | | |
| Bin | 43702.33 | 9 | 4855.81 | 62.88 | **<0.001** | 0.81 | 18313.4 | 9 | 2034.82 | 20.19 | **<0.001** | 0.61 |
| Error | 10434.53 | 135 | 77.22 | | | | 11788.35 | 117 | 100.76 | | | |
| TMS*bin | 278.1 | 18 | 15.45 | 1.83 | **0.02** | 0.11 | 45.94 | 18 | 2.55 | 0.17 | 0.99 | 0.01 |
| Error | 2280.41 | 270 | 8.45 | | | | 3420.56 | 234 | 14.62 | | | |
| UPPER ARM | SS | DOF | MS | F | P | Partial η² | SS | DOF | MS | F | p | Partial η² |
| TMS | 48.04 | 2 | 24.02 | 0.33 | 0.72 | 0.02 | 137.08 | 2 | 68.54 | 0.76 | 0.48 | 0.05 |
| Error | 2183.12 | 30 | 72.77 | | | | 2359.79 | 26 | 90.76 | | | |
| Bin | 22718.16 | 9 | 2524.24 | 54.33 | **<0.001** | 0.78 | 11297.79 | 9 | 1255.31 | 22.86 | **<0.001** | 0.64 |
| Error | 6271.85 | 135 | 46.46 | | | | 6426.21 | 117 | 54.92 | | | |
| TMS*bin | 318.95 | 18 | 17.72 | 1.80 | **0.03** | 0.11 | 66.84 | 18 | 3.71 | 0.28 | 0.99 | 0.02 |
| Error | 2664.26 | 270 | 9.87 | | | | 3148.37 | 234 | 13.45 | | | |

*TMS* main effect of TMS; Bin = main effect of time bin, *TMS*bin* interactive effect of TMS and time bin, *SS* sum of squares, *MS* mean square, *DOF* degrees of freedom.
Significant *p* values are marked in bold.
Index, wrist, forearm and upper arm represent the results obtained from the trajectories of these markers. For the readers' convenience, all posthocs' *p* of the main effect of the time bin were not reported in the text.

external force. This result seems unexpected, because V6A also contains proprioceptive cells[23,32], and the external force applied in the study exerted a proprioceptive perturbation on the arm. Moreover, our data is also not in agreement with a recent paper that found that late corrections of a reach-to-grasp movement elicited by the application of an external force are impaired after TMS over hV6A[29]. One reason for this apparent discordance could go beyond the sensory domain. In our task, the spatial position of the reaching target shifted during visual perturbations, forcing hV6A to update the spatial representation at the service of reaching. In contrast, during proprioceptive perturbations, the visual target remains stationary and the spatial representation invariant. This result parallels single neuron data in macaques highlighting a special role for V6A in spatial computations for reaching[10,14,33,34]; thus, hV6A could be causally involved in the updating of the spatial representation of the reaching target more than in a basic sensory-driven motor reprogramming, in keeping with previous results[31]. Additionally, the shift of the visual target, essential for hV6A, triggers also a shift of covert spatial attention, which modulates hV6A activity[31,35,36]. So, the impairments after hV6A stimulation could also be due to an impairment of attention. Alternatively, the confirmation of the role of hV6A in reach reprogramming specifically triggered by visual shifts, already found in one of our recent studies[5], reinforces the idea that hV6A primarily processes visual feedback to support motor control. Indeed, impairing hV6A computations by TMS induces later corrections of reaching.

## hPEc is involved in tardive reaching reprogramming driven by proprioceptive information

We found here that hPEc was specifically involved in reaching corrections driven by proprioceptive perturbations. This new finding is in keeping with the high incidence of limb somatosensory (mostly proprioceptive) responses found in macaque PEc, higher than that found in area V6A[21,22,25]. The TMS effect excluded the most distal segment of the arm, the index finger. This lack of effects on index finger might be due to a combination of arm/wrist movements that participants performed to compensate for the TMS effects. This compensation may have allowed them to maintain the index finger in a position that ensured high accuracy, thereby minimizing the potential influence of TMS stimulation on it. The absence of TMS effects during visual perturbations agrees with the weaker effect (compared to the nearby V6A) of visual input in PEc[25].

No other studies stimulated hPEc with the coordinates we used here. Nevertheless, in a recent TMS study, Marigold and coworkers[37], targeted a region slightly anterior to our hPEc site. In their study, Marigold et al provided the stimulation during a movement that featured a target jump very similar to our visual perturbation condition. They reported no effects on movement trajectories following stimulation of this area, in agreement with our results. This supports the hypothesis that hPEc might play a primary role in integrating proprioceptive feedback to support visually guided reaching corrections.

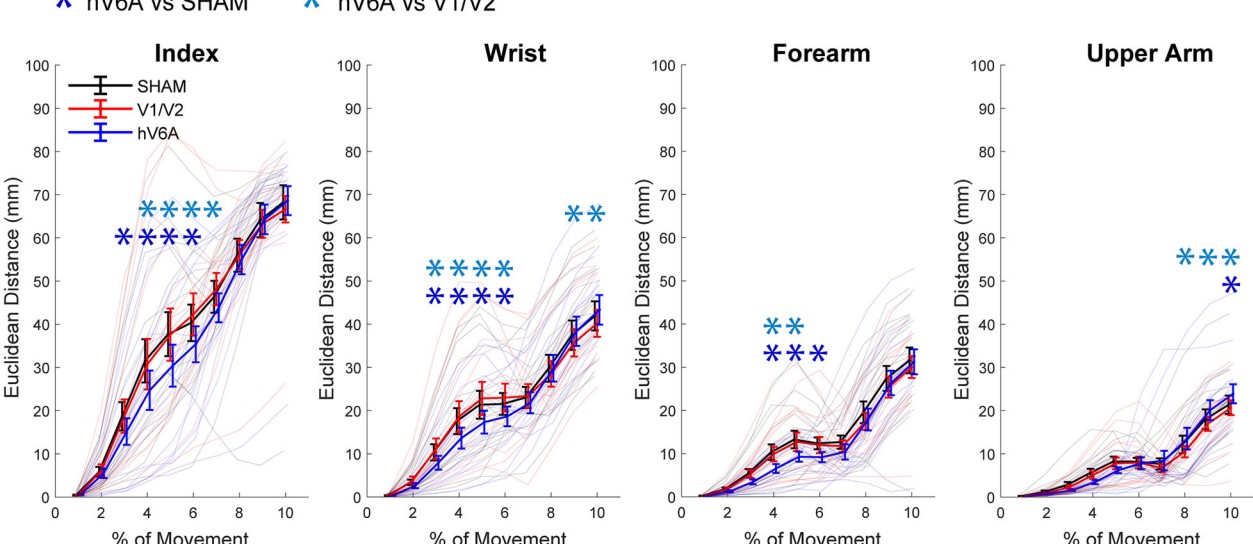

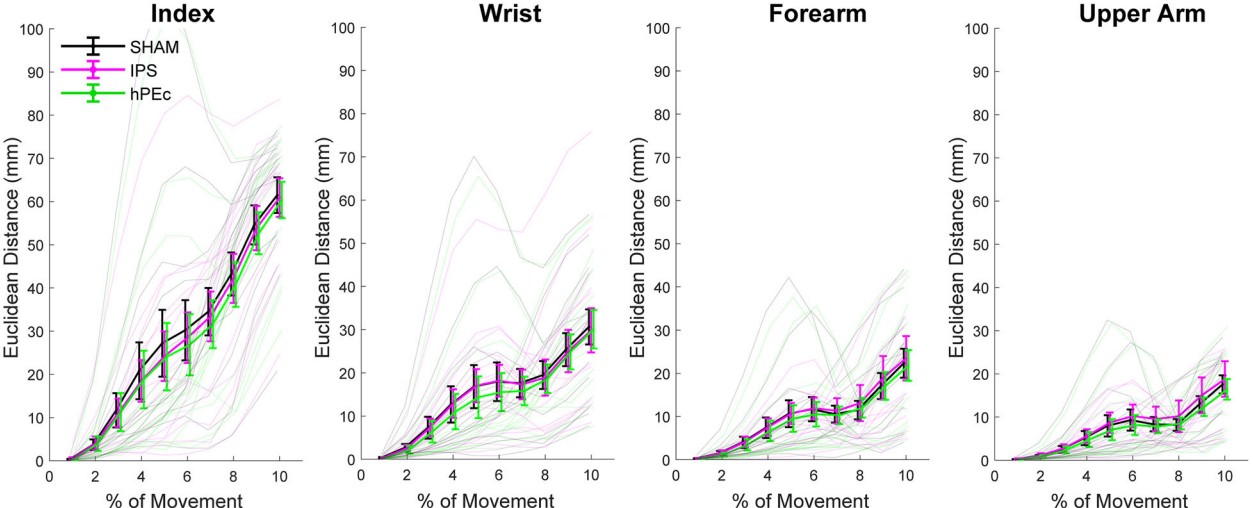

**Fig. 3 | ED results during visual perturbations. A** Results of Experiment 1; **B** Results of Experiment 2. ED at each time bin between trajectories from the unperturbed near condition and the visual perturbation condition in each stimulation condition. From the left to the right, results regarding the index, wrist, forearm and upper arm. Colors represent stimulation conditions (black = SHAM, Blue = hV6A, red = V1/V2, magenta = IPS, green = hPEc). Error bars represent ± one SEM. The asterisks mark a statistically significant post-hoc difference (Blue = hV6A vs SHAM, Light blue = hV6A vs V1/V2, Turquoise = V1/V2 vs SHAM; Light green = hPEc vs SHAM, Dark green = hPEc vs IPS, Moss green = IPS vs SHAM). Background lines indicate individual participants' data. In B, no significant differences between stimulation conditions were found.

Overall, we did not observe any significant effect on final reaching accuracy following hV6A or hPEc stimulation (see Supplementary material). This may be due to the timing of our TMS stimulation, which occurred during the early phase of the reaching movement. This likely gave participants sufficient time to recover from the TMS-induced effects and accurately reach the target, a benefit further supported by the continuous visibility of the target on the touchscreen throughout the trial.

The lack of effects on reaching accuracy after hV6A stimulation is not in keeping with the report of impairments in accuracy after stimulating anterior SPOC (hV6A)[38]. The reason for this discrepancy could be due to differences in the task design. Specifically, while in our study participants were stimulated during visually guided reaching execution, in the study

conducted by Vesia and colleagues the TMS was delivered during memory-guided reach planning. Given these differences, we consider the two studies complementary, both supporting a causal role of hV6A in the spatial encoding of reach targets.

### TMS effects in the occipital and parietal control sites
The stimulation of our active control sites (IPS and occipital cortex) showed significant effects. The impairments of corrections driven by proprioceptive perturbations after IPS stimulation align with those reported in a study[39], where IPS stimulation resulted in a prolonged deceleration phase of reaching corrections driven by an external force. These findings are also consistent with those of Della-Maggiore et al[40]., who reported similar effects

**Table 2 | ANOVA results for the ED during proprioceptive perturbations**

| | PROPRIOCEPTIVE PERTURBATION | | | | | | | | | | |
| | EXPERIMENT 1 | | | | | | EXPERIMENT 2 | | | | |
| INDEX | SS | DOF | MS | F | p | Partial η² | SS | DOF | MS | F | p | Partial η² |
|---|---|---|---|---|---|---|---|---|---|---|---|---|
| TMS | 392.2 | 2 | 196.1 | 1.35 | 0.27 | 0.08 | 155.4 | 2 | 77.7 | 0.60 | 0.553632 | 0.04 |
| Error | 4327.8 | 30 | 144.3 | | | | 339.2 | 26 | 128.4 | | | |
| Bin | 90853.9 | 9 | 10094.9 | 33.93 | **<0.001** | 0.69 | 96112.2 | 9 | 10679.1 | 30.12 | **<0.001** | 0.70 |
| Error | 4327.8 | 30 | 144.3 | | | | 41472.6 | 117 | 354.5 | | | |
| TMS*bin | 670.2 | 18 | 37.2 | 1.82 | **0.02** | 0.11 | 327.3 | 18 | 18.2 | 0.59 | 0.899436 | 0.04 |
| Error | 4327.8 | 30 | 144.3 | | | | 7115.6 | 234 | 30.4 | | | |
| WRIST | SS | DOF | MS | F | p | Partial η² | SS | DOF | MS | F | p | Partial η² |
| TMS | 75.9 | 2 | 37.9 | 1.16 | 0.33 | 0.07 | 71.7 | 2 | 35.8 | 0.70 | 0.50 | 0.05 |
| Error | 975.9 | 30 | 32.5 | | | | 1314.3 | 26 | 50.6 | | | |
| Bin | 48147.5 | 9 | 5349.7 | 30.24 | **<0.001** | 0.67 | 65427.7 | 9 | 7269.7 | 18.26 | **<0.001** | 0.58 |
| Error | 23877.5 | 135 | 176.9 | | | | 46571 | 117 | 398 | | | |
| TMS*bin | 232 | 18 | 12.9 | 1.18 | 0.27 | 0.07 | 751.4 | 18 | 41.7 | 3.19 | **<0.001** | 0.20 |
| Error | 2925.3 | 270 | 10.8 | | | | 3055.7 | 234 | 13.1 | | | |
| FOREARM | SS | DOF | MS | F | p | Partial η² | SS | DOF | MS | F | p | Partial η² |
| TMS | 135.94 | 2 | 67.97 | 1.99 | 0.15 | 0.12 | 63 | 2 | 31.5 | 0.43 | 0.65 | 0.03 |
| Error | 1022.81 | 30 | 34.09 | | | | 1884.2 | 26 | 72.5 | | | |
| Bin | 37907.22 | 9 | 4211.91 | 27.41 | **<0.001** | 0.65 | 57591.7 | 9 | 6399.1 | 18.78 | **<0.001** | 0.60 |
| Error | 20743.33 | 1135 | 153.65 | | | | 39865.1 | 117 | 340.7 | | | |
| TMS*bin | 138.57 | 18 | 7.7 | 0.80 | 0.69 | 0.05 | 472.1 | 18 | 26.2 | 2.42 | **<0.002** | 0.16 |
| Error | 2593.43 | 270 | 9.61 | | | | 2528.4 | 234 | 10.8 | | | |
| UPPER ARM | SS | DOF | MS | F | p | Partial η² | SS | DOF | MS | F | p | Partial η² |
| TMS | 62.8 | 2 | 31.4 | 2.44 | 0.10 | 0.14 | 64.1 | 2 | 32 | 0.48 | 0.62 | 0.04 |
| Error | 385.6 | 30 | 12.85 | | | | 1721 | 26 | 66.2 | | | |
| Bin | 24232.55 | 9 | 2692.51 | 25.61 | **<0.001** | 0.63 | 44193.5 | 9 | 4910.4 | 19.51 | **<0.001** | 0.60 |
| Error | 14189.97 | 135 | 105.11 | | | | 29440 | 117 | 251.6 | | | |
| TMS*bin | 76.07 | 18 | 4.23 | 0.79 | 0.70 | 0.05 | 453.8 | 18 | 25.2 | 2.80 | **<0.001** | 0.18 |
| Error | 1427.5 | 270 | 5.29 | | | | 2106.9 | 234 | 9 | | | |

Same conventions as in Table 1. Significant p values are marked in bold.

when stimulating an area in the left PPC roughly corresponding to our IPS stimulation site.

Interestingly, the effects we found after TMS over a lateral site (IPS) during proprioceptive perturbations occurred earlier than those observed after stimulation of a more medial site (hPEc), suggesting that the contribution of hPEc in reaching reprogramming was dependent on IPS. This perfectly aligns with the concept of dorsomedial dependence on dorsolateral visual stream activity developed by Verhagen[41] in the control of grasping movements, and extends this concept to the control of reaching corrections.

Regarding the occipital cortex, the impairments of proprioceptive corrections found here were also observed in another similar study[40]. Unexpectedly, the effects observed here after occipital stimulation in proprioceptive corrections were not also seen during visual perturbations. The lack of TMS visual masking effect during visual perturbations could be because the part of the visual field represented in our occipital stimulation site (the central one) is likely not crossed by the arm during reaching. Moreover, a recent study[42] pointed out that the part of V1 representing the central visual field was more activated for somatosensory than for visual exploration of objects, even if they were located peripherally. So, the specific, apparently surprising effects of TMS over V1/V2 during proprioceptive perturbations suggest that the occipital lobe is involved in the somatosensory processing that may affect the capability to correct reaching movements. In further agreement with this, other studies[43,44] found that during

motor planning, information about the intended movement can be decoded from early visual cortex activity. Thus, several findings align with ours and suggest a possible, though still far from being fully understood, role of V1 in proprioceptive processing at the service of motor control. It is also important to remember that TMS effects, while site-specific, are not necessarily confined to the targeted area. TMS can in fact modulate neural activity both locally and in functionally connected regions[45–47]. Therefore, an alternative explanation for our findings is that TMS-induced excitation targeting V1/V2 may have propagated through interconnected networks, potentially influencing areas involved in proprioceptive feedback processing during visually guided reaching tasks.

## Conclusions

In this study, we demonstrate that the human PPC, in parallel with macaque studies, is essential for real-time reach control and is organized into subregions with distinct functional roles. Our results reveal that hV6A played a causal role in early reaching corrections only when the target position had changed due to a visual shift, whereas hPEc and IPS were involved in corrections driven by the application of an external force with a later time course. These findings suggest that the PPC engages distinct neural circuits to meet the specific demands of accurate reaching control under different perturbations. These results enhance our understanding of the neural mechanisms governing sensorimotor integration during online adjustments of goal-directed actions. By

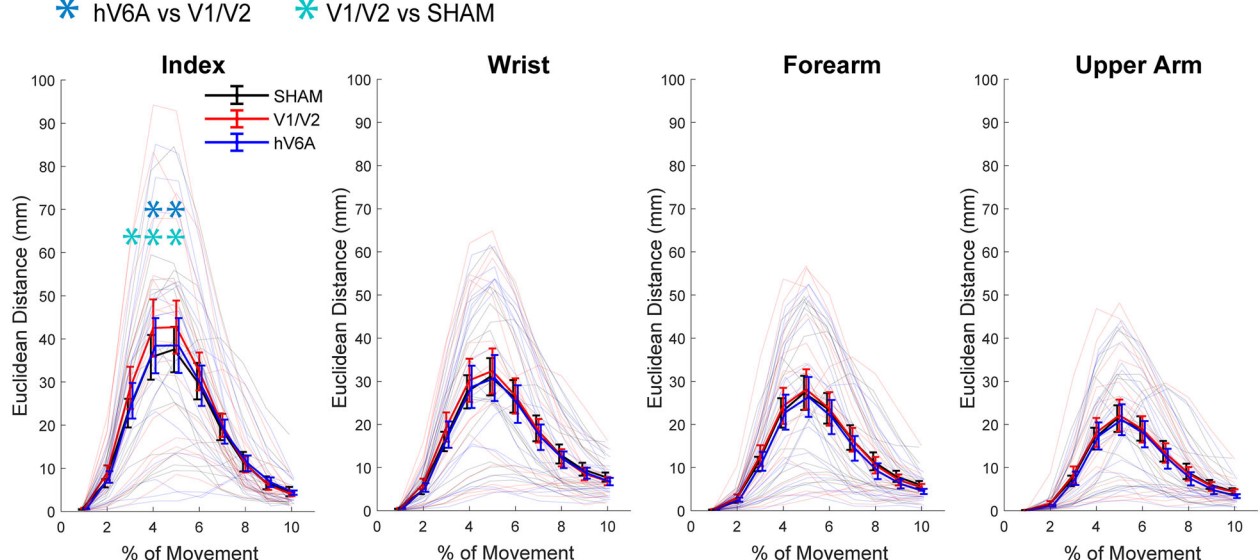

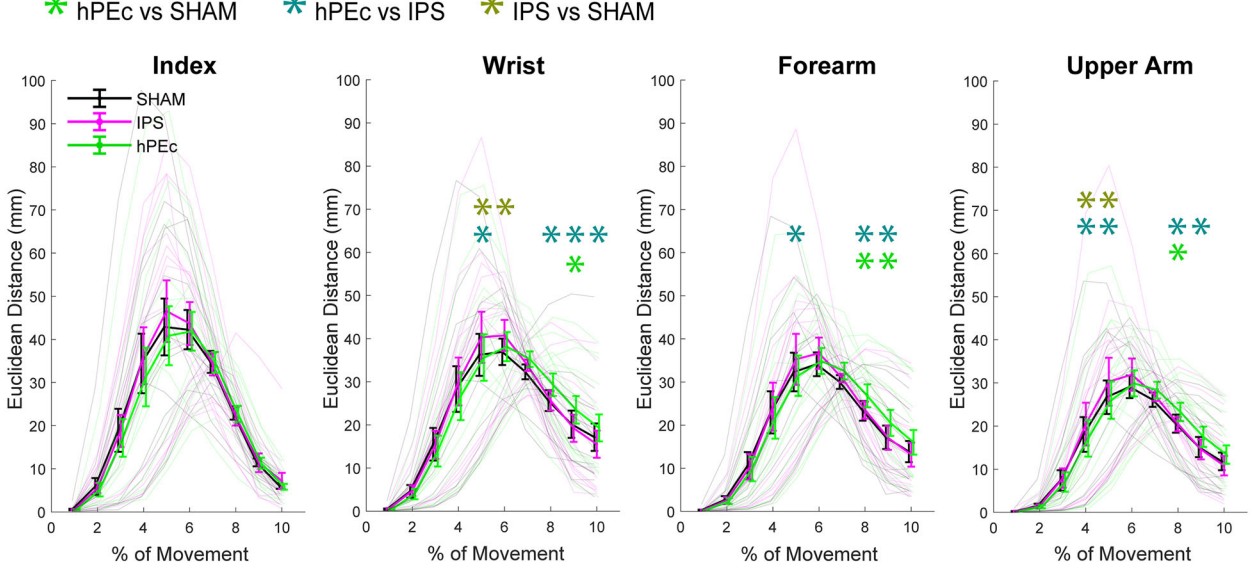

**Fig. 4 | Euclidean distances results for the proprioceptive perturbations. A** Results of Experiment 1; **B** Results of Experiment 2. ED at each time bin between trajectories from the unperturbed far condition and the proprioceptive perturbation condition for each stimulation site. All other conventions as in Fig. 3.

uncovering these neural mechanisms, our work fills a critical gap in the understanding of sensorimotor integration for motor control. We believe these deeper insights are pivotal for advancing neurorehabilitation and neural prosthetics, especially in a society that increasingly relies on neuro-technologies to drive clinical breakthroughs.

## Methods

### Participants

Thirty healthy adult participants were involved in this study. They were divided into two groups: one group of sixteen took part in the first experiment (average age 23.69 ± 3.77, age range: 19-32, 2 males); another group of fourteen took part in the second experiment (average age 24.86 ± 3.46, age range: 20-32, 5 males). The two experiments were different only regarding the areas stimulated by TMS (see TMS protocol). The participants were

classified as right-handed based on the Edinburgh Handedness Inventory[48] and had normal or corrected-to-normal visual acuity in both eyes. Participants provided written informed consent, and the procedures were approved by the Bioethical Committee at the University of Bologna (Prot. 0057635) and were in accordance with the ethical standards of the 2013 Declaration of Helsinki. No discomfort or adverse effects during TMS were reported or noticed.

### Apparatus and experimental design

Stimuli were displayed on a 19-inch touchscreen (ELO IntelliTouch 1939 L) set horizontally on a desk in front of the participants. For stimuli presentation, we used Matlab (Mathworks, Natick, MA, USA, RRID:SCR_001622) with the Psychtoolbox extension[49]. Participants were seated on a chair in a darkened room, with their head stabilized by a head/

chin rest to minimize head movements. In all trials, the reaching movement started with the participant's right hand on a button placed on the desk (home button) centered on the body midline, centered to the head position. A purpose-built, Matlab-controlled, noiseless electrical winch (pulling machine) was used to pull a thin wire connected to the participant's wrist applying a defined force-time profile.

Each trial started with the home button pressing. Upon pressing, a green dot (diameter 0.4°) appeared straight ahead on the screen (Fig. 2A-B) serving as a fixation point. Participants were instructed to maintain their gaze on the fixation point until the task ended. After a variable period of 1.3–1.5 s, a gray target (diameter 0.4°) appeared to the right of the fixation point, at either 11° ('near' target) or 20° of eccentricity ('far' target, Fig. 2B). The onset of the gray target served as a go signal, prompting participants to initiate a reaching movement toward it. They were instructed to move their hand at a fast yet comfortable speed, while maintaining accuracy when touching the target with the tip of the index finger. The task included two types of trials: unperturbed (80%) and perturbed (20%). This 80%-20% trial distribution was intentionally designed to ensure that reaching perturbations were genuinely unexpected, a standard approach in psychophysical research (see[50–54]), where rare events (typically occurring in less than 25% of trials) are perceived as unexpected by participants. In unperturbed trials, the target remained stationary (near target, 40% of trials, far target, 40% of trials). There were two types of perturbed trials, both requiring rightward online reaching corrections. In 10% of trials, the correction was triggered by an unexpected shift of the visual target (from now on called 'visual perturbations', Fig. 2A). In these trials, the target initially appeared in the near position and shifted to the far location 50 ms after movement onset, requiring a rightward movement to reach the target. In another 10% of trials, the target remained stable in the far position, but after 50 ms from movement onset, the pulling machine pulled the participant's wrist in the leftward direction with a force of 8 N (0 ms rise time, 80 ms peak duration, 100 ms fall time). Thus, participants had to correct the movement by applying a rightward force to reach the far target (from now on these trials are called 'proprioceptive perturbations', Fig. 2A). In this way, the required reach correction after either visual or proprioceptive perturbation was in the same direction. In all unperturbed and perturbed trials, TMS pulses were delivered during movement execution, with the first pulse released 50 ms after the movement onset (Fig. 2C).

Each block of both experiments consisted of 60 randomized trials (24 unperturbed near, 24 unperturbed far, 6 visually perturbed, and 6 proprioceptively perturbed). Each block was repeated twice for each stimulation condition (SHAM, V1/V2, hV6A, counterbalanced for Experiment 1, and SHAM, IPS, hPEc, counterbalanced for Experiment 2, Fig. 1A-B), for a total of 360 trials in a single experimental session. Each session lasted approximately 3 hours, with participants always using their right arm.

## TMS protocol: localization of brain sites and stimulation

The coil position on each participant's scalp was determined before each experimental session through a neuronavigation procedure (Cortexplore, Linz, Austria)[5,55,56]. In the initial phase, anatomical landmarks on the scalp of each participant (nasion, inion, bilateral preauricular points, and vertex) were recorded. Following, the nasion-inion line was traced, and 65 additional points were collected to provide a uniform representation of the subject-specific scalp surface. Coordinates in Talairach space were automatically estimated by the Cortexplore Navigator, from an MRI-constructed stereotaxic template in each participant, and the scalp sites were marked on it with a surgical pen.

In both experiments, we tested two active stimulation sites: a region of interest (ROI) and a control area. In Experiment 1, the ROI was left hV6A, while the active control area was the bilateral occipital cortex (from now on called 'V1/V2', Fig. 1A). In Experiment 2, the ROI was the left hPEc with the left IPS as the active control site (Fig. 1B). Additionally, both experiments included a SHAM condition.

To target left hV6A, we used Talairach coordinates (x = −10, y = −78, z = 40) that have been consistently used in our previous studies[5,35,57,58]. For

bilateral V1/V2, the coil was positioned 2 cm above the inion to achieve bilateral stimulation[35,57,59,60]. To target IPS, we used Talairach coordinates x = −44, y = −39, z = 51 (converted from the MNI coordinates of aIPS[39]: x = −44, y = −42, z = 55), and, for hPEc, the Talairach coordinates were x = −13, y = −57, z = 55 (converted from the MNI coordinates of [20]: x = −13, y = −61, z = 62, where activations of hPEc were observed for arm movements). All ethical regulations relevant to human research participants were followed. The selection of the IPS as an active control site in experiment 2 instead of somatosensory cortex was motivated by the focus of our study on the functional specialization of distinct subregions within the PPC, and at the same time by the necessity to avoid motor twitches which could be observed if targeting somatosensory cortex with suprathreshold intensity. SHAM stimulation was performed by placing the coil tilted at 90° over the vertex bilaterally, so that participants could feel coil–scalp contact and discharge noise as during active stimulation, but no current was induced in the brain[61,62]. Electric field (E-field) modeling of our TMS stimulations (Fig. 1, left) was performed using the open source software SimNIBS[63] and visualized through the software Gmsh[64].

Biphasic TMS pulses (10 Hz, 3 pulses, as performed in other studies on the medial PPC[5,38,65], Fig. 2C) were delivered using a Deymed DuoMAG XT (DEYMED, Hronov, Czech Republic) stimulator connected to a 70 mm figure-of-eight coil. TMS pulses (which ranged from 0 to 200 ms after the perturbation onset) were administered for the entire period in which the processing for initiating the corrections occurred. Stimulation of hV6A, hPEc and V1/V2 was carried out by placing the coil tangentially over the scalp site along a parasagittal line with the handle pointing downward[5,38,57]. IPS stimulation was performed by placing the coil tangentially over the scalp and tilting it 45° from the tangential line[66]. The coil was held by a counter-balanced coil holder which, together with the head/chin rest, minimized the head movements during the task. This absence of relative coil-head movements was confirmed by checking the coil position after each block of trials.

To set TMS intensity, the resting motor threshold (rMT) was estimated for all participants using standard procedures[62] (see Supplementary material). For all stimulation sites, the intensity of magnetic stimulation was fixed at 120% of the rMT, as in Ref.[5]. The rMTs of the two groups of participants were not statistically different (two-sample t-test, p = 0.37). In the first experiment, the range of intensities was 50-77% (mean 60.13 ± 7.78) of the total stimulator output, while in the second experiment was 53-77% (mean 62.83 ± 7.85). No phosphenes were perceived by the participants.

## Kinematics and electromyographic (EMG) recordings

The kinematics of reaching movements were recorded using a motion tracking system (VICON motion capture system, 6 M cameras, 1024 × 1024 pixel resolution) by sampling the position of four markers at a frequency of 100 Hz. Markers were attached on the nail of the index finger (reaching finger), on the wrist (on the radial styloid process), on the midpoint of the forearm, and on the midpoint of the upper arm (Fig. 5A). We recorded entire upper limb kinematics because, during the reach-to-point movements in our task design, different segments of the upper limb (upper arm, forearm, wrist, and index finger) contributed with varying degrees of freedom. During each stimulation session, EMG was used to monitor muscle activity of the extensor carpi radialis (ECR) and of the flexor carpi radialis (FCR) muscles (Fig. 5B and Fig. S1-2). Kinematic data were processed by interpolating the trajectories to 100 time points, effectively expressing each movement as a percentage of its total duration (Fig. 5C–F). This interpolation preserved the overall shapes of the trajectories, ensuring that the temporal dynamics remain consistent, and the normalization allowed us to analyze and interpret when specific effects occur during the movement, enabling more meaningful and consistent comparisons across trials, independent of absolute movement time.

## Ethics approval

All procedures were approved by the Bioethical Committee at the University of Bologna (Prot. 0057635) and were in accordance with the ethical standards of the 2013 Declaration of Helsinki.

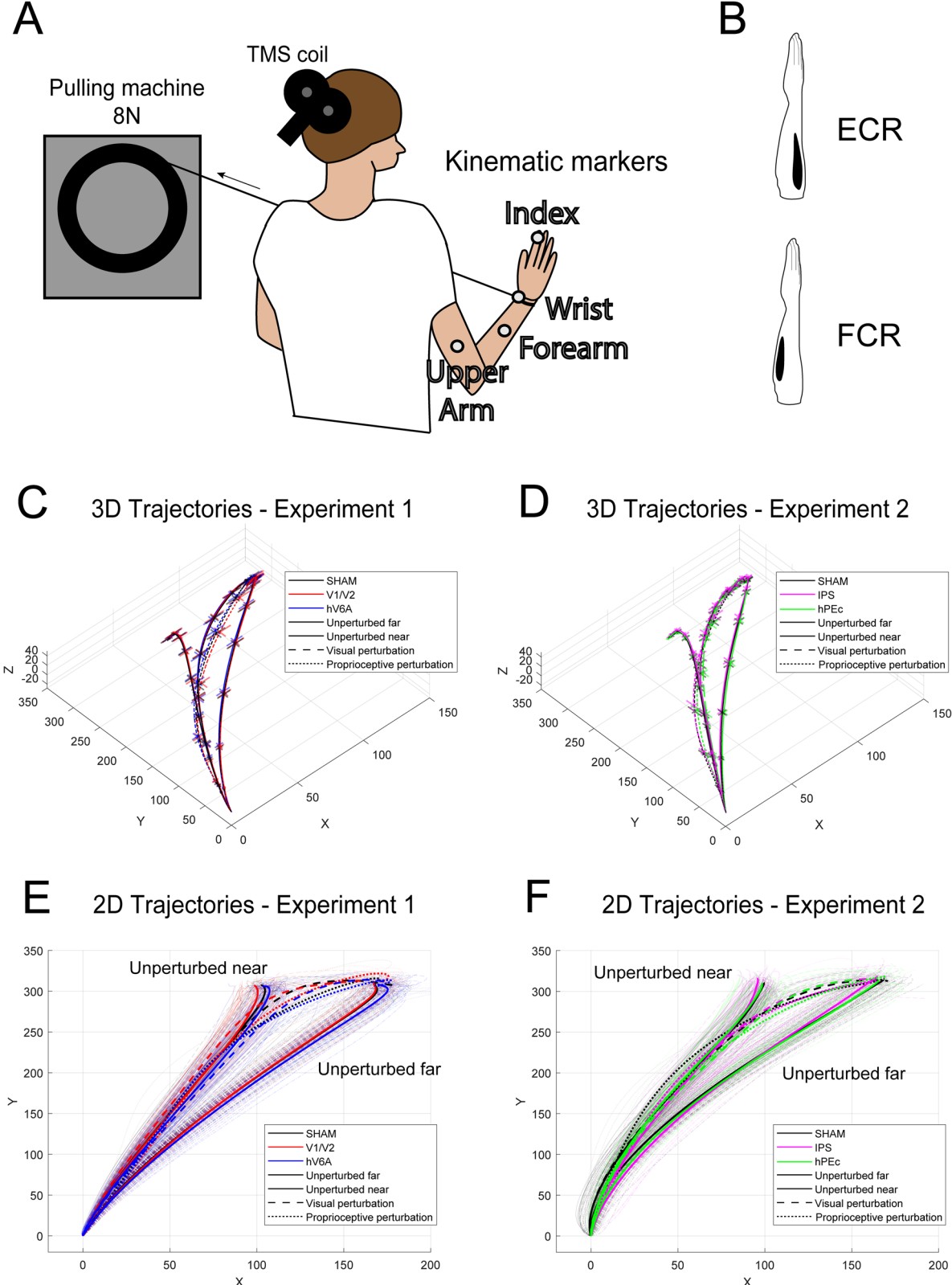

**Fig. 5 | Experimental apparatus for proprioceptive perturbations and EMG recordings. A** The pulling machine was attached to the participant's wrist and pulled the arm with a force of 8 N leftward. Four kinematic markers were attached at different positions on the right hand, forearm and upper arm. **B** Representation of the location of the extensor carpi radialis (ECR) and flexor carpi radialis (FCR) muscles from which EMG activity was recorded. **C, D** Three-dimensional plots showing the average index finger trajectories across participants for each experiment and stimulation condition. Error bars represent ± one SEM. **E, F** Two-dimensional plots showing the single-trial index finger trajectories (thin lines) along with their averages (thick lines) of two representative participants (one for each experiment) under each stimulation condition. The trajectories of the wrist, forearm and upper arm are shown in the Supplementary materials (Figs. S5-6).

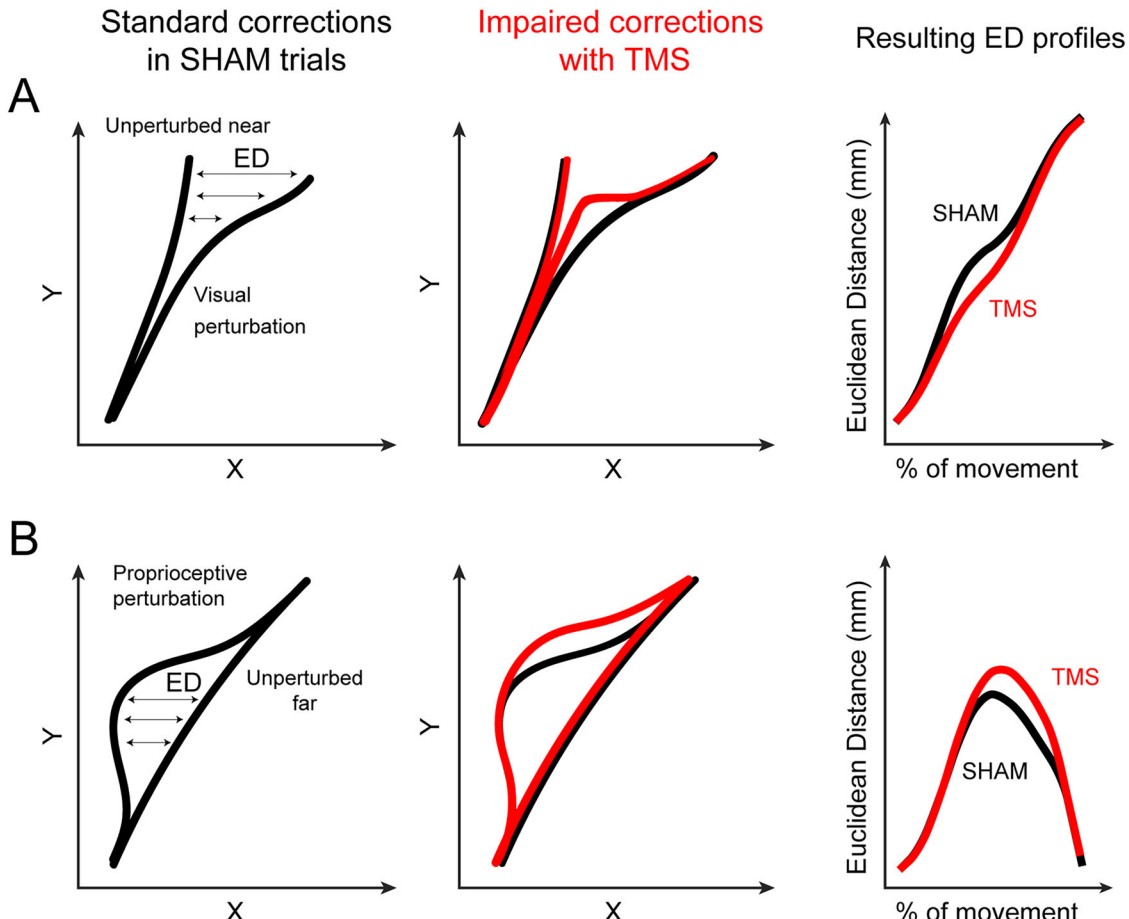

**Fig. 6 | Calculation of the Euclidean distance (ED). A** The corrective ability during visual perturbations was evaluated using the trajectory points of the perturbed trajectory and the unperturbed 'near' one. After TMS, we expected an impaired corrective response of the reach, leading to a lower ED. **B** The corrective ability during proprioceptive perturbations was estimated by using the perturbed trajectory and the unperturbed 'far' one, and after TMS we expected a higher ED. For clarity, trajectories are shown in 2D (x and y), although the ED was calculated in 3D (x, y, z). Black: example trajectories during Sham; red: example of impaired trajectories after TMS (expected results); % of movement=% of movement time.

**Reporting summary**. Further information on research design is available in the Nature Portfolio Reporting Summary linked to this article.

## Statistics and reproducibility
### Analysis of reaching corrections
We measured the reaching correction capability by calculating, as previously done[5], the 3D Euclidean distance (ED) on each participant's average trajectories. Since the trajectories were recorded in milliseconds and varied in length, we first interpolated each to 100 points (expressed as % of the movement) and then binned them into 10 bins (each one representing 10% of the movement). For each pair of normalized trajectories, the ED was calculated at each data point as follows:

$$ED_i = \sqrt[2]{(x_{cui} - x_{cp/ui})^2 + (y_{cui} - y_{cp/ui})^2 + (z_{cui} - z_{cp/ui})^2}$$

Where x, y and z are, respectively, the horizontal, sagittal and elevation component of the trajectories, c is the stimulation condition, p/u are the perturbed or unperturbed conditions and i is the $i^{th}$ time bin of the trajectory. We performed this analysis for each kinematic marker. We wanted to subtract the perturbed trajectories from the unperturbed ones with the same initial trajectory. So, during visual perturbations, participants initially reached toward the same target as in the near unperturbed condition, then adjusted their movement following the visual shift of the target to the far one. Thus, in this case, we chose as the unperturbed trajectory the near one

(Fig. 6A). During proprioceptive perturbations, participants began the reaching movement toward the far target but then their arm was pulled leftward, and they corrected by applying a force rightward towards the far target. So, in this case, we chose as the unperturbed trajectory the far one (Fig. 6B). We measured ED for each stimulation condition, looking for significant changes in ED following TMS. The effects of TMS were assessed by a 2-way repeated measures analysis of variance (ANOVA) on the ED (3 levels, SHAM, V1/V2, hV6A for Experiment 1 and SHAM, IPS, hPEc for Experiment 2) and bin (10 levels, bin 1-10); we did this three times: for unperturbed trials, for visual perturbations and for proprioceptive perturbations. In both experiments, keeping different EDs over time between each stimulation condition and SHAM was informative of a different capability to correct the movement (Figs. 3–4). We also calculated the ED between the two unperturbed conditions (Fig. S4).

### Other analyses
The analysis of EMG responses (Fig. S1-2), movement times (Fig. S3), reaction times and reaching accuracy are reported in the Supplementary material.

In all the analyses, post-hoc tests were carried out with the Newman-Keuls correction for multiple comparisons.

### Reproducibility
Our sample size consisted of thirty healthy adult participants. They were divided into two groups: one group of sixteen took part in the first

**Article**

experiment (average age 23.69 ± 3.77, age range: 19-32, 2 males); another group of fourteen took part in the second experiment (average age 24.86 ± 3.46, age range: 20-32, 5 males).

## Data availability

The datasets generated and/or analyzed during the current study are not publicly available because the informed consent signed by the volunteers enrolled in the study did not contain the possibility to share the data publicly. Nevertheless, data is available from the corresponding author upon reasonable request. Numerical source data underlying the figures can be found in Supplementary Data.

## Code availability

This study used standard, custom-built MATLAB programmed scripts that are available from the lead contact upon request.

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

## Acknowledgements

This study was carried out thanks to: the Space It Up project funded by the Italian Space Agency, ASI, and the Ministry of University and Research, MUR, under contract n. 2024-5-E.0 - CUP n. I53D24000060005; the Project PRIN20208RB4N9 by MUR; the funding from Fondazione Cassa di Risparmio in Bologna (CARISBO); the funding from the European Union's Horizon 2020 research and innovation programme under the Future and Emerging Technologies grant agreement No. 951910- MAIA; the project "Multisensory integration of locomotion-related visual and somatomotor signals- MulWALK", codice proposta: 2022BK2NPS_001 - CUP: J53D23010900006 Finanziato dall'Unione Europea - NextGenerationEU a valere sul Piano Nazionale di Ripresa e Resilienza (PNRR) – Missione 4 Istruzione e ricerca – Componente 2 Dalla ricerca all'impresa - Investimento 1.1, Avviso Prin2022 indetto con DD N. 104 del 2/2/2022; the funding by #NEXTGENERATIONEU (NGEU) and funded by the Ministry of University and Research (MUR), National Recovery and Resilience Plan (NRRP), project MNESYS (PE0000006) – A Multiscale integrated approach to the study of the nervous system in health and disease (DN. 1553 11.10.2022). Authors wish to thank the participants and the undergraduate students for proficient support to the experimental sessions and analyses. Authors also wish to thank Claire Corbeaux for verifying the manuscript's English language proficiency.

## Author contributions

Ro.B., C.G. and P.F. designed research; Ri.B., Ro.B. and D.D.G set up the lab and performed research; Ri.B. and Ro.B analyzed data; Ri.B. wrote the first draft of the paper; Ro.B. and P.F. acquired funding; Ro.B. supervised the research. All authors revised the paper. All authors have seen and approved the final version of the manuscript.

## Competing interests

The authors declare no competing interests.

## Consent to participate

Informed consent was obtained from all participants.
