## [Transparent Peer Review file · Communications Biology]

Functional Specialization of the Human Posterior Parietal Cortex in Visually and Proprioceptively Driven Reaching Corrections

Corresponding Author: Professor Rossella Breveglieri

This manuscript has been previously submitted at another journal. This document only contains information relating to versions considered at Communications Biology.

Version 0:

Reviewer comments:

Reviewer #1

(Remarks to the Author)

Summary and Evaluation

This is an outstanding study that provides clear and compelling evidence for functional dissociations within the human posterior parietal cortex during online motor corrections. Using well-timed, perturbation-triggered rTMS, the authors demonstrate that hV6A is selectively involved in visually guided reach corrections, while hPEc supports proprioceptive-based adjustments. These findings offer novel insights into the distinct roles of PPC subregions in feedback-driven motor control.

The study is exceptionally well designed and executed. The authors employ rigorous controls, including active control sites and sham stimulation, and their analytic approach is thorough and appropriate. The interpretations are thoughtful, clearly argued, and well supported by the data, with consideration of alternative explanations. The findings are well integrated into the existing literature and represent a significant and meaningful contribution to the field of sensorimotor neuroscience.

Minor Considerations and Suggestions

1. Discussion structure: To improve clarity, consider beginning the Discussion section with a concise summary of the main findings before delving into detailed interpretation and implications.
2. Line 206: It would strengthen the discussion to explicitly relate the current findings to those of Vesia et al. (2010), who used rTMS to target SPOC during visually guided reaching. Discussing these results would help position hV6A's role within the broader visuomotor network.
3. Line 256: Consider elaborating on how potential effects from V1 might reflect its role within a distributed network, rather than isolated early visual processing. This could clarify possible indirect contributions to the observed behavioral outcomes.
4. Figure suggestion: Including example trials from one representative subject, showing reach trajectories across conditions, would help readers visualize the behavioral effects and complement the group-level results

Reviewer #2

(Remarks to the Author)

review of Breveglieri et al, Functional Specialization of the Human Posterior Parietal Cortex in Visually and Proprioceptively Driven Reaching Corrections

The authors applied repetitive transcranial magnetic stimulation (rTMS) over two parietal areas, the proposed human homologues of monkey areas V6A and PEc, while human participants executed reaches to visual targets. In some trials, the target jumped (visual perturbation) or the arm was bumped (proprioceptive perturbation). rTMS Stimulation of hV6A (h indicating human) affected reach corrections to visual but not proprioceptive reach corrections elicited by the respective perturbations. Conversely, rTMS stimulation of hPEc affected reach corrections elicited by proprioceptive but not visual perturbations. The authors interpret these findings as indicating that human parietal cortex houses functionally distinct subregions for motor control.

I find this a well-written paper with clear, adequate methodology and very interesting and relevant results. Below, I will discuss some aspects that, in my view, would improve the paper further.
Respectfully submitted, signed, Tobias Heed

Given the format of "Results first, Methods last", I suggest a few changes to the manuscript. I think Results should be rewritten to better lay open the design. It was not clear to me in the beginning that the two critical regions were stimulated between-subjects - but this is important information. For instance, the results report around line 163 seem to suggest that ALL conditions are within-subject, as V6A and PEc are reported side by side without ever mentioning that these are different subject pools and ANOVAs. Related, I suggest moving Figure 3 to the Results, as it is vital to understand the paper (task and stimulated regions). If it is placed there, then explaining that two pools of subjects were run "comes natural" anyway. Moreover, it would be good to emphasize that the experiment tested visual vs. proprioceptive perturbations within-subject (and even within-blocks, which I find particularly convincing methodology, as it avoids participants forming modality-specific expectations in different phases of the experiment). Finally, I suggest to clarify the term rTMS early on. Many different protocols fall under the term "rTMS", and it is helpful for the reader to know that TMS, despite it being "r", occurred trial by trial. (This, too, automatically comes along with moving Fig. 3).

I am missing information on final reach accuracy. It seems implicit in the report that participants compensated for TMS, but it is not reported. Please add. It would also be good if the authors could discuss this point. If final accuracy is adequate for visual and proprio perturbations despite TMS, does this imply further corrections up along the motor processing network, or recovery of V6A/PEc from the TMS stimulation?

Fig. 1 and 2: I find the insertions about significance confusing and difficult to read. It may be a better option to leave out the exact numbers and just place different-colored stars (for different compared conditions) at the exact time points at which the difference was significant.

line 167, it seems as if the statistical tests don't catch what is really going on for hPEc. It looks as if the effect here is that responses are delayed, which is not caught by testing each time point separately. This is relevant for two reasons: 1, the non-significant index finger (Fig. 2, lower row) still seems to show a delay. 2, The unexpected effect of V1/V2 is not shifted, so seems to express something different than the effect elicited by hPEc. Maybe there is a way to estimate the maxima of the reach curves in Fig. 2 to tickle this out better?

I find the argument in line 218 for the index finger not showing an effect implausible (also see my previous comment about an apparent delay). Even if PEc were to control only the arm but not the fingers, changes in finger position would follow changes of arm position, unless a different region can control finger position independently of arm control - which seems implausible. It seems to me that the null result for the finger here is much more likely to be due to chance or the specific combinations of arm/wrist movements that happened to occur.

typos etc.

line 31, visual or proprioceptive - both was done, so "and", not "or"

p4, first header, "role of..."

p4, Table 1, third line, $p = 0.00$ is not sensible - typo?

line 107, "segment involved" - this is ambiguous, because the reader doesn't know whether perturbations were somehow planned to affect a specific segment (which "involved" suggests) or whether the text is just reporting the effects of the same perturbation on different parts of the arm

line 121 or => and

line 126 and Fig. 1 "arm" => "upper arm"

line 154, distalmost one => the most distal one

line 200, unclear what a "focal reason" is; Moreover, the reasoning is not clear to me. How was stimulation applied in ref 30 to go along with what is outlined here? This should be explained more clearly.

line 216 stimulations => perturbations

Reviewer #3

(Remarks to the Author)

The manuscript by Riccardo et al. entitled, "Functional Specialization of the Human Posterior Parietal Cortex in Visually and Proprioceptively Driven Reaching Corrections" investigated the role of parietal areas in online reaching coordination. Using a three-pulse transcranial magnetic stimulation sequence to human V6A (hV6A) and PEc (hPEc), compared to sham stim at different brain sites, they tested participant responses to visual or mechanical perturbations during reaching movements. The data lead the authors to conclude that hV6A has a causal role in response to visual perturbations and hPEc has a causal role in response to proprioceptive or mechanical perturbations. The manuscript is well written, the experiments are well thought-out, and the topic is highly important to the field. There are some areas of concern that I suggest addressing.

Major concerns:

1. Given that the type of perturbation was entirely dependent on the near-far target attribution, participants were likely able to infer the type of perturbation coming up based on the original target location, which could have influenced their anticipatory strategies and potentially undermined the internal validity of the task manipulation.

2. Subject-specific MRIs were not used for neuronavigation, raising concerns about the accuracy of stimulation targeting. Relatedly, based on Fig 6 in Pitzalis et al., 2019, the hPEc location seems to be on the boundary between regions that process foot-pointing and hand-pointing. Given that ventral side of PPC was more involved in hand-pointing, shouldn't the hPEc associated to wrist proprioception be more to the ventral? It would benefit the paper if the authors could justify the hPEc coordinates, provide detailed descriptions of neuronavigation procedures including whether neuronavigational targeting was subject-specific, how head positioning was monitored throughout the study, and preferably, electric field (E-field) modeling to highlight the focality of TMS.

3. It is not clear why aIPS was selected as an active control for the hPEc site. For example, if primary visual cortex serves as a control for hV6A, then wouldn't the analogous control site for hPEc would likely be S1/3?

4. The rationale for analyzing the entire upper limb kinematics in a simple reach-to-point task is unclear. The authors do not discuss joint segment differences or provide justification for analyzing kinematics beyond the endpoint (e.g., index finger) measures. It is not evident what additional insight this broader kinematic data provides in terms of testing hypotheses or explaining mechanisms.

5. The authors describe that the rise time for mechanical force perturbation is 0 ms. Is this physiologically realistic? It seems that it would take some time for the mechanical perturbation to generate force on the limb (e.g., due to mass, inertia, etc.). Moreover, it is not clear if the duration of the perturbation is similar across perturbation types. For example, the discontinuation of a force perturbation acts as a second perturbation in a way, and effect that does not occur with visual perturbations.

6. The EMG data seems to have been clipped at the time of TMS delivery, which is 50ms after movement onset. It would be helpful to show the EMG data prior to movement onset to illustrate that the signals did not differ until TMS.

7. The Euclidean distance outcome measure is a very nice measure, however there are two concerns. First, the measure is analyzed as % of movement time because there were substantial differences in movement times between the conditions. While it is understandable why the normalization is done, it does nevertheless introduce a distortion in the shape of the Euclidean traces making it difficult to know whether between-condition differences are real or an artifact of stretched profiles. Relatedly, it would help very much to see raw data traces in the paper prior to diving into Euclidean distance measures to understand the kinematics.

Minor comments

1. Line 337: "hPEc".

2. Line 343: The MNI coordinates are not specified for hPEc.

3. Table 1: the p value of index TMS*bin is 0. Additionally, DOF is incomplete as it is missing values for error.

4. Using "stimulation conditions" to indicate different brain sites is a bit confusing. Perhaps Please using 'brain site' would be better.

5. Figs 1 and 2: Lines showing the significant difference between stimulation sites are difficult to see which bin reached the significant level.

Version 1:

Reviewer comments:

Reviewer #2

(Remarks to the Author)

Thank you for considering my feedback to the previous draft and revising the paper accordingly, and congratulations for this very interesting and informative study. I only have a few language-related suggestions. Best, Tobias Heed

line 115, I suggest calling the trial types slightly differently. "perturbed visual trials" sounds as if participants were doing a "visual reach", i.e. had a visual target. But what is meant is that the perturbation was visual => "visually perturbed trials" (and analogue for proprio)

line 135, factor Bin is not really clear - it is not mentioned before, and should be briefly explained here. Similarly, factor TMS could be more clearly specified, e.g. as TMS location or as TMS (levels: location1, location2, sham)

line 136 have been => were

line 200, remove "higher" (?)

line 212, distalmost => most distal

Reviewer #3

(Remarks to the Author)

The authors have sufficiently addressed my concerns. No further concerns are noted.

Reviewer #1

Summary and Evaluation

This is an outstanding study that provides clear and compelling evidence for functional dissociations within the human posterior parietal cortex during online motor corrections. Using well-timed, perturbation-triggered rTMS, the authors demonstrate that hV6A is selectively involved in visually guided reach corrections, while hPEc supports proprioceptive-based adjustments. These findings offer novel insights into the distinct roles of PPC subregions in feedback-driven motor control.

The study is exceptionally well designed and executed. The authors employ rigorous controls, including active control sites and sham stimulation, and their analytic approach is thorough and appropriate. The interpretations are thoughtful, clearly argued, and well supported by the data, with consideration of alternative explanations. The findings are well integrated into the existing literature and represent a significant and meaningful contribution to the field of sensorimotor neuroscience.

REPLY: We deeply thank the Reviewer for the positive comments and for the time spent in reviewing the paper.

Minor Considerations and Suggestions

1. Discussion structure: To improve clarity, consider beginning the Discussion section with a concise summary of the main findings before delving into detailed interpretation and implications.

REPLY: We agree that adding an introductory paragraph enhances the clarity of our manuscript. In accordance with the Reviewer's suggestion, we have included it at the beginning of the revised Discussion.

2. Line 206: It would strengthen the discussion to explicitly relate the current findings to those of Vesia et al. (2010), who used rTMS to target SPOC during visually guided reaching. Discussing these results would help position hV6A's role within the broader visuomotor network.

REPLY: Accordingly, we believe that the comparison with the results of (Vesia et al., 2010) could enrich the Discussion. Thus, we have inserted it in a dedicated paragraph at lines 290-296 in the revised Discussion. We there emphasize how these studies are complementary in their findings. We thank the Reviewer for this valuable suggestion.

3. Line 256: Consider elaborating on how potential effects from V1 might reflect its role within a distributed network, rather than isolated early visual processing. This could clarify possible indirect contributions to the observed behavioral outcomes.

REPLY: We agree with the Reviewer's observation. As is well established, TMS effects, while site-specific, are not necessarily confined to the targeted area. Indeed, TMS can influence not only local neural activity but also remote, functionally connected regions (Avenanti et al., 2013; Siebner et al., 2009; Valchev et al., 2015). Therefore, it is possible to think that TMS-induced effect targeting V1/V2 may have propagated through connected networks, potentially reaching areas involved in proprioceptive feedback processing during visually guided reaching tasks. We added this consideration in the revised text at lines 324-329.

4. Figure suggestion: Including example trials from one representative subject, showing reach trajectories across conditions, would help readers visualize the behavioral effects and complement the group-level results

REPLY: As requested, we have updated the new Figure 5 and add a new supplementary figure to include trajectory data from a representative participant from each experiment. We added in figure 5 data from the index finger, while in the supplementary data (Fig. S5) data from the wrist, forearm and upper arm markers of a representative participant for each experiment. Moreover, to complement group-level results, we have inserted the data points of individual participants in the graphs of Figures 3-4.

Reviewer #2:

review of Breveglieri et al, Functional Specialization of the Human Posterior Parietal Cortex in Visually and Proprioceptively Driven Reaching Corrections

The authors applied repetitive transcranial magnetic stimulation (rTMS) over two parietal areas, the proposed human homologues of monkey areas V6A and PEc, while human participants executed reaches to visual targets. In some trials, the target jumped (visual perturbation) or the arm was bumped (proprioceptive perturbation). rTMS Stimulation of hV6A (h indicating human) affected reach corrections to visual but not proprioceptive reach corrections elicited by the respective perturbations. Converseley, rTMS stimulation of hPEc affected reach corrections elicited by proprioceptive but not visual perturbations. The authors interpret these findings as indicating that human parietal cortex houses functionally distinct subregions for motor control.

I find this a well-written paper with clear, adequate methodology and very interesting and relevant results. Below, I will discuss some aspects that, in my view, would improve the paper further.

Respectfully submitted, signed, Tobias Heed

REPLY: We thank the Reviewer for appreciating our work and for the time spent in reviewing the paper.

Given the format of "Results first, Methods last", I suggest a few changes to the manuscript.

I think Results should be rewritten to better lay open the design. It was not clear to me in the beginning that the two critical regions were stimulated between-subjects - but this is important information. For instance, the results report around line 163 seem to suggest that ALL conditions are within-subject, as V6A and PEc are reported side by side without ever mentioning that these are different subject pools and ANOVAs. Related, I suggest moving Figure 3 to the Results, as it is vital to understand the paper (task and stimulated regions). If it is placed there, then explaining that two pools of subjects were run "comes natural" anyway. Moreover, it would be good to emphasize that the experiment tested visual vs. proprioceptive perturbations within-subject (and even within-blocks, which I find particularly convincing methodology, as it avoids participants forming modality-specific expectations in different phases of the experiment).

REPLY: We appreciate the Reviewer's suggestion and fully understand the importance of providing a clearer introduction to the brain regions stimulated and the task design before presenting the results. As requested, we have added two introductory paragraphs at the beginning of the Results section: the first outlines the brain sites targeted in each experiment, and the second summarizes the task design. Additionally, we have introduced a new Figure 1, which now includes both the stimulated brain regions and the newly added electric field (E-field) modeling (in response to Reviewer 3). According to the Reviewer's request, the revised Figure 3 (now Figure 2), which illustrates the task design, has been repositioned into the Results accordingly to follow this updated structure. We also emphasize that each experiment tested visual vs. proprioceptive perturbations within-subject, as requested (Lines 88-117).

Finally, I suggest to clarify the term rTMS early on. Many different protocols fall under the term "rTMS", and it is helpful for the reader to know that TMS, despite it being "r", occurred trial by trial. (This, too, automatically comes along with moving Fig. 3).

REPLY: Done (see line 89). See also the reply to the previous point. Thank you.

I am missing information on final reach accuracy. It seems implicit in the report that participants compensated for TMS, but it is not reported. Please add. It would also be good if the authors could discuss this point. If final accuracy is adequate for visual and proprio perturbations despite TMS, does this imply further corrections up along the motor processing network, or recovery of V6A/PEc from the TMS stimulation?

REPLY: The information regarding reaching accuracy is briefly mentioned in line 229-230 of the main text and further detailed in the supplementary material, where we report that no significant effects of TMS were observed. We acknowledge the necessity of further discuss these results. We believe this outcome might be in fact related to the timing of the TMS stimulation in our task design. Specifically, the stimulation was applied early, likely affecting the initial phase of the movement, this choice was made to specifically target the online correction of reaching movement. It is plausible that participants were able to "recover" from the TMS interference by the end of the movement. This interpretation is also supported by the Euclidean distance analyses of the index finger kinematic marker (the final effector of our reaching movements), which consistently showed no significant differences between conditions in the final bins of the movement.

We now address this point in the revised discussion (lines 285-289), including also a comparison with the results from Vesia's work (Vesia et al., 2010), which we believe is highly relevant to this topic and in agreement also with the response to point 2 of Reviewer 1.

Fig. 1 and 2: I find the insertions about significance confusing and difficult to read. It may be a better option to leave out the exact numbers and just place different-colored stars (for different compared conditions) at the exact time points at which the difference was significant.

REPLY: Thank you for the suggestion. We agree that the clarity of the figures can be improved by adding colored stars corresponding to the bins with significant differences. We have revised the figures accordingly.

line 167, it seems as if the statistical tests don't catch what is really going on for hPEc. It looks as if the effect here is that responses are delayed, which is not caught by testing each time point separately. This is relevant for two reasons: 1, the non-significant index finger (Fig. 2, lower row) still seems to show a delay. 2, The unexpected effect of V1/V2 is not shifted, so seems to express something different than the effect elicited by hPEc. Maybe there is a way to estimate the maxima of the reach curves in Fig. 2 to tickle this out better?

REPLY: We agree with the Reviewer so, to address this point, we performed the following analysis as requested: for each marker, subject and stimulation condition (SHAM, IPS, hPEc), we extracted the Euclidean distance profiles and computed the maximum value of each curve. We then took at which time point this maximum value was occurring, this resulted in a distribution of time points for each condition. We then performed a one-way ANOVA with TMS condition as the factor to assess potential differences across conditions. The analysis revealed no significant effects (index finger: $F = 2.7$, partial $\eta^2 = 0.17$, $p = 0.09$; wrist: $F = 0.8$, partial $\eta^2 = 0.06$, $p = 0.46$; forearm: $F = 1.2$, partial $\eta^2 = 0.08$, $p = 0.32$; upper arm: $F = 1$, partial $\eta^2 = 0.07$, $p = 0.38$). We understand that, based on visual inspection of the curves, there may appear to be a potential effect. However, this trend does not translate into statistical significance, so we think these results are not informative enough to be included in the revised manuscript.

I find the argument in line 218 for the index finger not showing an effect implausible (also see my previous comment about an apparent delay). Even if PEc were to control only the arm but not the fingers, changes in finger position would follow changes of arm position, unless a different region can control finger position independently of arm control - which seems implausible. It seems to me that the null result for the finger here is much more likely to be due to chance or the specific combinations of arm/wrist movements that happened to occur.

REPLY: We acknowledge that our initial speculation based on electrophysiological data might appear limited, and we agree with the Reviewer that the absence of that effect might be attributed to a coordination mechanism instead. Given the multiple degrees of freedom available in the arm during reaching movements in our paradigm, it is plausible that participants compensated for any perturbations by adjusting their arm, forearm, or wrist. This may have allowed them to maintain the index finger (the primary effector for reaching) in a position that ensured high accuracy, thereby minimizing the potential influence of TMS stimulation on it. We rephrased the concept previously expressed at this regard, adding these new considerations in the revised discussion (lines 272-276).

typos etc.

line 31, visual or proprioceptive - both was done, so "and", not "or"

REPLY: Fixed, thank you.

p4, first header, "role of..."

REPLY: Fixed, thank you.

p4, Table 1, third line, $p = 0.00$ is not sensible - typo?

REPLY: Fixed with <0.001 , thank you.

line 107, "segment involved" - this is ambiguous, because the reader doesn't know whether perturbations were somehow planned to affect a specific segment (which "involved" suggests) or whether the text is just reporting the effects of the same perturbation on different parts of the arm

REPLY: We rephrased accordingly, thank you.

line 121 or => and

REPLY: Fixed, thank you.

line 126 and Fig. 1 "arm" => "upper arm"

REPLY: Fixed, thank you.

line 154, distalmost one => the most distal one

REPLY: Fixed, thank you.

line 200, unclear what a "focal reason" is; Moreover, the reasoning is not clear to me. How was stimulation applied in ref 30 to go along with what is outlined here? This should be explained more clearly.

REPLY: we understand the confusion of the Reviewer about our phrasing. We have rephrased this point to be clearer about the discordance between our data in the paper cited.

line 216 stimulations => perturbations

REPLY: Fixed, thank you.

Reviewer #3:

The manuscript by Riccardo et al. entitled, "Functional Specialization of the Human Posterior Parietal Cortex in Visually and Proprioceptively Driven Reaching Corrections" investigated the role of parietal areas in online reaching coordination. Using a three-pulse transcranial magnetic stimulation sequence to human V6A (hV6A) and PEc (hPEc), compared to sham stim at different brain sites, they tested participant responses to visual or mechanical perturbations during reaching movements. The data lead the authors to conclude that hV6A has a causal role in response to visual perturbations and hPEc has a causal role in response to proprioceptive or mechanical perturbations. The manuscript is well written, the experiments are well thought-out, and the topic is highly important to the field. There are some areas of concern that I suggest addressing.

REPLY: Thank you for these suggestions and considerations, and for the time spent in reviewing the paper.

Major concerns:

1. Given that the type of perturbation was entirely dependent on the near-far target attribution, participants were likely able to infer the type of perturbation coming up based on the original target

location, which could have influenced their anticipatory strategies and potentially undermined the internal validity of the task manipulation.

REPLY: Even though we understand the rationale behind this concern, we believe our task design did not present this caveat. Specifically, as addressed in the Methods section, unperturbed trials constituted the majority of the experiment (80%), while perturbed trials accounted for only 20%. This distribution supports the conclusion that reaching perturbations were genuinely unexpected, a common design choice in psychophysical research (see (Brandolani et al., 2025; Paulignan et al., 1991; Pisella et al., 2000; Posner, 1980; Rumiati & Humphreys, 1998), where unexpected events (typically under 25%) are used to ensure participants perceive them as surprising. Additionally, all conditions were fully randomized within each block. This design choice was intentional, aimed at preventing participants from anticipating any specific perturbation, as Reviewer 2 also noticed in their point 1. We rephrased the methods section of our manuscript to better express these concepts at lines 374-377.

2. Subject-specific MRIs were not used for neuronavigation, raising concerns about the accuracy of stimulation targeting. Relatedly, based on Fig 6 in Pitzalis et al., 2019, the hPEc location seems to be on the boundary between regions that process foot-pointing and hand-pointing. Given that ventral side of PPC was more involved in hand-pointing, shouldn't the hPEc associated to wrist proprioception be more to the ventral? It would benefit the paper if the authors could justify the hPEc coordinates, provide detailed descriptions of neuronavigation procedures including whether neuronavigational targeting was subject-specific, how head positioning was monitored throughout the study, and preferably, electric field (E-field) modeling to highlight the focality of TMS.

REPLY: Unfortunately, we were unable to have the structural individual MRIs because of the absence of any MRI scanner for research at the University of Bologna. As a result, we opted for an MRI-constructed stereotaxic template. This method, already validated in previous studies (Breveglieri, Brandolani, Diomedi, et al., 2025; Breveglieri, Brandolani, Galletti, et al., 2025; Fleischmann et al., 2020; Riis et al., 2025; Turrini et al., 2025), provided a reliable substitute for MRI-based targeting.

Regarding the hPEc coordinates, we have used the Talairach-converted coordinates of coordinates of Pitzalis et al. (2019) (Pitzalis et al., 2019) to target the human homologue of the macaque area PEc. While this area shows activation related to leg movements, it also exhibits activity associated with hand movements (Pitzalis et al., 2019). Thus, we expected to find hand-related effects by targeting these coordinates. As we used the same coordinates of the Pitzalis's study, we were also able to complement with TMS the fMRI results regarding hand activations in human PEc. We have added the justification in line 417 of the revised paper.

We appreciate the Reviewer's suggestion to clarify the neuronavigation procedure. In the revision, we have now included a more detailed description of the procedure (lines 400-403), together with the requested electric field (E-field) modeling of our TMS stimulation sites and with the head position monitoring (line 434-36). The E-field models are presented in a new figure (new Figure 1) that has been added to the revised manuscript.

3. It is not clear why aIPS was selected as an active control for the hPEc site. For example, if primary visual cortex serves as a control for hV6A, then wouldn't the analogous control site for hPEc would likely be S1/3?

REPLY: The decision to use IPS as a control site was based on several considerations. First, S1 is proximal to M1, and this makes it theoretically highly sensitive and prone to eliciting motor responses (Holmes et al., 2019). These motor twitches, likely elicitable with our stimulation intensity (3 pulses of TMS at 120% of the resting motor threshold) would have added noise during the reaching movements that could have confounded our results. It should be noted, relatedly, that the stimulation intensity of TMS over S1 is usually subthreshold (Vidoni et al., 2010). Performing rTMS at 120% of the motor threshold over S1 could thus have compromised the integrity of our data and the aims of our experiments.

Said this, as our study focused on the functional specialization of different subregions within the posterior parietal cortex, we considered IPS a more appropriate control. We added a better explanation of our rationale at lines 418-421.

4. The rationale for analyzing the entire upper limb kinematics in a simple reach-to-point task is unclear. The authors do not discuss joint segment differences or provide justification for analyzing kinematics beyond the endpoint (e.g., index finger) measures. It is not evident what additional insight this broader kinematic data provides in terms of testing hypotheses or explaining mechanisms.

REPLY: In our task, the reaching movement started from a position close to the subject's chest (see Figure 2), ending to a point quite far from it, involving a reaching movement that required the entire arm to be engaged. So, we thought that a more detailed kinematic analysis of different segments of the arm would be more appropriate. Additionally, during a reach-to-point movement, different segments of the upper limb, such as the arm, forearm, wrist, and index finger, contribute with varying degrees of freedom. We have included this explanation in the revised discussion and in the methods (see lines 449-451).

5. The authors describe that the rise time for mechanical force perturbation is 0 ms. Is this physiologically realistic? It seems that it would take some time for the mechanical perturbation to generate force on the limb (e.g., due to mass, inertia, etc.). Moreover, it is not clear if the duration of the perturbation is similar across perturbation types. For example, the discontinuation of a force perturbation acts as a second perturbation in a way, and effect that does not occur with visual perturbations.

REPLY: We have not performed a formal characterization of the rise time (time to go from 10% to 90% of the programmed force), so what we can do is stating that the system was built to minimize it and reduce it to a value well below any "humanly relevant" time: i) The motor's current control loop has a bandwidth of 2.5 kHz. Then, the current rise time can be estimated with $T_r = 0.35 / BW = 0.14$ ms. ii) The system operates in torque control mode, with torque linearly proportional to current and output shaft's angular velocity being almost zero. iii) The controller-motor system is very powerful, delivering a peak power of 3 kW. iv) The inertia of the transmission system has been minimized by using an optimized 3D-printed hollow plastic pulley and a rigid nylon cable. Thus, we can estimate the force rise time with $10 * T_r = 1.4$ ms, virtually zero.

Regarding the perturbations, we validated their duration by analyzing movement times (see Supplementary Data). Our analysis showed no significant differences in movement time between visual and proprioceptive perturbations in both experiment 1 and 2, suggesting that the two types of perturbations were comparable in duration.

6. The EMG data seems to have been clipped at the time of TMS delivery, which is 50ms after movement onset. It would be helpful to show the EMG data prior to movement onset to illustrate that the signals did not differ until TMS.

REPLY: As requested, we have now inserted in the revised Supplementary materials of the manuscript the analysis of the EMG data prior to movement onset. No differences were observed.

7. The Euclidean distance outcome measure is a very nice measure, however there are two concerns. First, the measure is analyzed as % of movement time because there were substantial differences in movement times between the conditions. While it is understandable why the normalization is done, it does nevertheless introduce a distortion in the shape of the Euclidean traces making it difficult to know whether between-condition differences are real or an artifact of stretched profiles. Relatedly, it would help very much to see raw data traces in the paper prior to diving into Euclidean distance measures to understand the kinematics.

REPLY: We thank the Reviewer for appreciating our choice of analysis. We chose to express our results as a percentage of the movement duration specifically to avoid potential misinterpretations occurring from variability in movement lengths. By normalizing the data in this way, reaching movements of different durations can be directly compared. This allows us to analyze and interpret at which percentage of the movement a given effect occurs, making comparisons across trials more meaningful and consistent, regardless of absolute movement time. The data were also interpolated to a uniform length across conditions to facilitate direct comparison. We understand the concern of the Reviewer, but this interpolation preserves the overall shape of the trajectories, ensuring that the temporal dynamics remain consistent. We understand that this might not have been explicated in the main text, so we added a better explanation of our analysis at lines (455-458).

We fully agree that including a visualization of the kinematic data would significantly enhance the reader's understanding of the dynamics of reaching movements across experimental conditions. In response to this suggestion, we have included the raw data in the figure 5 and Supplementary figures S5-6. We hope this addition improves the accessibility and interpretability of the kinematic process.

Minor comments

1. Line 337: "hPEc".

REPLY: Fixed, thank you.

2. Line 343: The MNI coordinates are not specified for hPEc.

REPLY: The Talairach coordinates for area hPEc in the manuscript (x = -13, y = -57, z = 55) were converted from the MNI coordinates of the same area, reported in (Pitzalis et al., 2019) and now included in the revised manuscript.

3. Table 1: the p value of index TMS*bin is 0. Additionally, DOF is incomplete as it is missing values for error.

REPLY: Fixed, thank you.

4. Using “stimulation conditions” to indicate different brain sites is a bit confusing. Perhaps Please using ‘brain site’ would be better.

REPLY: We fixed some spots in which using “brain sites” was more appropriate, however when recalling the SHAM stimulation we kept “stimulation conditions” since “SHAM” is not technically a brain site.

5. Figs 1 and 2: Lines showing the significant difference between stimulation sites are difficult to see which bin reached the significant level.

REPLY: We agree and fixed this in the revised versions of the figures.

References:

- Avenanti, A., Annella, L., Candidi, M., Urgesi, C., & Aglioti, S. M. (2013). Compensatory Plasticity in the Action Observation Network: Virtual Lesions of STS Enhance Anticipatory Simulation of Seen Actions. *Cerebral Cortex*, 23(3), 570–580. <https://doi.org/10.1093/cercor/bhs040>
- Brandolani, R., Galletti, C., Fattori, P., Breveglieri, R., & Poletti, M. (2025). Distinct modulation of microsaccades in motor planning and covert attention. *Scientific Reports*, 15(1), 19580. <https://doi.org/10.1038/s41598-025-03000-z>
- Breviglieri, R., Brandolani, R., Diomedì, S., Lappe, M., Galletti, C., & Fattori, P. (2025). Role of the Medial Posterior Parietal Cortex in Orchestrating Attention and Reaching. *The Journal of Neuroscience*, 45(1), e0659242024. <https://doi.org/10.1523/JNEUROSCI.0659-24.2024>
- Breviglieri, R., Brandolani, R., Galletti, C., Avenanti, A., & Fattori, P. (2025). Time-dependent enhancement of corticospinal excitability during cortico-cortical paired associative stimulation of the hV6A-M1 network in the human brain. *NeuroImage*, 316, 121301. <https://doi.org/10.1016/j.neuroimage.2025.121301>
- Fleischmann, R., Köhn, A., Tränkner, S., Brandt, S. A., & Schmidt, S. (2020). Individualized Template MRI Is a Valid and Reliable Alternative to Individual MRI for Spatial Tracking in Navigated TMS Studies in Healthy Subjects. *Frontiers in Human Neuroscience*, 14. <https://doi.org/10.3389/fnhum.2020.00174>
- Holmes, N. P., Tamè, L., Beeching, P., Medford, M., Rakova, M., Stuart, A., & Zeni, S. (2019). Locating primary somatosensory cortex in human brain stimulation studies: experimental evidence. *Journal of Neurophysiology*, 121(1), 336–344. <https://doi.org/10.1152/jn.00641.2018>
- Paulignan, Y., MacKenzie, C., Marteniuk, R., & Jeannerod, M. (1991). Selective perturbation of visual input during prehension movements. *Experimental Brain Research*, 83(3). <https://doi.org/10.1007/BF00229827>

- Pisella, L., Gréa, H., Tilikete, C., Vighetto, A., Desmurget, M., Rode, G., Boisson, D., & Rossetti, Y. (2000). An “automatic pilot” for the hand in human posterior parietal cortex: Toward reinterpreting optic ataxia. *Nature Neuroscience*, *3*(7), 729–736. <https://doi.org/10.1038/76694>
- Pitzalis, S., Serra, C., Sulpizio, V., Di Marco, S., Fattori, P., Galati, G., & Galletti, C. (2019). A putative human homologue of the macaque area PEc. *NeuroImage*. <https://doi.org/10.1016/j.neuroimage.2019.116092>
- Posner, M. I. (1980). Orienting of attention. *The Quarterly Journal of Experimental Psychology*, *32*(1), 3–25., *32*(1), 3–25. <https://doi.org/10.1080/00335558008248231>
- Riis, T. S., Lunt, S., & Kubanek, J. (2025). MRI free targeting of deep brain structures based on facial landmarks. *Brain Stimulation*, *18*(1), 131–137. <https://doi.org/10.1016/j.brs.2024.12.1478>
- Rumiati, R. I., & Humphreys, G. W. (1998). Recognition by action: Dissociating visual and semantic routes to action in normal observers. *Journal of Experimental Psychology: Human Perception and Performance*, *24*(2), 631–647. <https://doi.org/10.1037//0096-1523.24.2.631>
- Siebner, H. R., Hartwigsen, G., Kassuba, T., & Rothwell, J. C. (2009). How does transcranial magnetic stimulation modify neuronal activity in the brain? Implications for studies of cognition. *Cortex*, *45*(9), 1035–1042. <https://doi.org/10.1016/j.cortex.2009.02.007>
- Turrini, S., Fiori, F., Arcara, G., Romei, V., di Pellegrino, G., & Avenanti, A. (2025). State-dependent associative plasticity highlights function-specific premotor-motor pathways crucial for arbitrary visuomotor mapping. *Science Advances*, *11*(20). <https://doi.org/10.1126/sciadv.adu4098>
- Valchev, N., Ćurčić-Blake, B., Renken, R. J., Avenanti, A., Keysers, C., Gazzola, V., & Maurits, N. M. (2015). cTBS delivered to the left somatosensory cortex changes its functional connectivity during rest. *NeuroImage*, *114*, 386–397. <https://doi.org/10.1016/j.neuroimage.2015.04.017>
- Vesia, M., Prime, S. L., Yan, X., Sergio, L. E., & Crawford, J. D. (2010). Specificity of human parietal saccade and reach regions during transcranial magnetic stimulation. *Journal of Neuroscience*, *30*(39), 13053–13065. <https://doi.org/10.1523/JNEUROSCI.1644-10.2010>
- Vidoni, E. D., Acerra, N. E., Dao, E., Meehan, S. K., & Boyd, L. A. (2010). Role of the primary somatosensory cortex in motor learning: An rTMS study. *Neurobiology of Learning and Memory*, *93*(4), 532–539. <https://doi.org/10.1016/j.nlm.2010.01.011>

Reviewer #2:

Thank you for considering my feedback to the previous draft and revising the paper accordingly, and congratulations for this very interesting and informative study. I only have a few language-related suggestions. Best, Tobias Heed

REPLY: We deeply thank the Reviewer!

line 115, I suggest calling the trial types slightly differently. "perturbed visual trials" sounds as if participants were doing a "visual reach", i.e. had a visual target. But what is meant is that the perturbation was visual => "visually perturbed trials" (and analogue for proprio)

REPLY: Done.

line 135, factor Bin is not really clear - it is not mentioned before, and should be briefly explained here. Similarly, factor TMS could be more clearly specified, e.g. as TMS location or as TMS (levels: location1, location2, sham).

REPLY: Done.

line 136 have been => were. **REPLY: Done.**

line 200, remove "higher" (?) **REPLY: Done.**

line 212, distalmost => most distal **REPLY: Done.**

Reviewer #3:

The authors have sufficiently addressed my concerns. No further concerns are noted.

REPLY: We deeply thank the Reviewer for the time spent on the revision.